# Pancreatic Cancer and Venous Thromboembolism

**DOI:** 10.3390/ijms25115661

**Published:** 2024-05-23

**Authors:** Teagan Prouse, Mohammad A. Mohammad, Sonali Ghosh, Narender Kumar, Ma. Lorena Duhaylungsod, Rinku Majumder, Samarpan Majumder

**Affiliations:** 1Department of Interdisciplinary Oncology, Louisiana State University Health Sciences Center, New Orleans, LA 70112, USA; tprous@lsuhsc.edu (T.P.); mmoha4@lsuhsc.edu (M.A.M.); sghos2@lsuhsc.edu (S.G.); nkuma1@lsuhsc.edu (N.K.); mduhay@lsuhsc.edu (M.L.D.); 2Department of Genetics, Louisiana State University Health Sciences Center, New Orleans, LA 70112, USA

**Keywords:** protein S (PS) 1, pancreatic ductal adenocarcinoma (PDAC), thrombosis

## Abstract

Pancreatic ductal adenocarcinoma (PDAC) accounts for more than 90% of all pancreatic cancers and is the most fatal of all cancers. The treatment response from combination chemotherapies is far from satisfactory and surgery remains the mainstay of curative strategies. These challenges warrant identifying effective treatments for combating this deadly cancer. PDAC tumor progression is associated with the robust activation of the coagulation system. Notably, cancer-associated thrombosis (CAT) is a significant risk factor in PDAC. CAT is a concept whereby cancer cells promote thromboembolism, primarily venous thromboembolism (VTE). Of all cancer types, PDAC is associated with the highest risk of developing VTE. Hypoxia in a PDAC tumor microenvironment also elevates thrombotic risk. Direct oral anticoagulants (DOACs) or low-molecular-weight heparin (LMWH) are used only as thromboprophylaxis in PDAC. However, a precision medicine approach is recommended to determine the precise dose and duration of thromboprophylaxis in clinical setting.

## 1. Introduction

Pancreatic ductal adenocarcinoma (PDAC) is currently the fourth leading cause of cancer-related deaths in the United States [1,2]. It is projected that, by 2030, it will become the second leading cause of cancer-related deaths [3]. Pancreatic ductal adenocarcinoma (PDAC) accounts for 90% of all pancreatic cancers (PCs), with a 5-year survival rate of less than 10% following diagnosis [4,5]. PDAC has the worst prognosis of any malignancy [6]. Notably, PDAC is linked to the highest rate of cancer-associated thrombosis (CAT) of all solid tumor malignancies [4,5]. Approximately 7.4% of patients diagnosed with PDAC will experience a form of VTE within 1 year of diagnosis, compared to overall cancer patients who have a risk of 3.1% [7]. A correlation between venous thromboembolism (VTE) and cancer was first reported by French physician Armand Trousseau in 1865 [8]. Since then, there has been an increasing awareness of the fundamental link between VTE and cancer. Within one year of cancer diagnosis, the development of VTE in a patient portends a poor prognosis [9]. Clinically, common manifestations of VTE primarily include pulmonary embolism (PE) and deep vein thrombosis (DVT), both of which are life-threatening conditions [10]. Other not so common appearances of VTE include superficial vein thrombosis, catheter-related thrombosis, and thrombi in unusual sites in the body [11]. In addition to the potentially fatal complications of thromboembolism (TE) events, cancer patients also report that thromboembolic events have a significant negative influence on overall quality of life [12]. Overall, the degree of cancer progression plays a key role in the risk of developing VTE [13].

The presence of metastasis within six months to two years after cancer diagnosis also significantly increases a patient’s risk of developing VTE [14,15]. Advanced tumor stage, increased age, genetic susceptibility, reduced mobility, various anticancer treatments, and the innate biological characteristics of the tumor further drive the development of VTE [16]. Some genetic risk factors for VTE include oncogenic mutations, such as the MET and KRAS mutations in solid tumors [17], the latter mutation being found in ~95% of pancreatic ductal adenocarcinoma (PDAC) solid tumors [18]. The activation of the coagulation cascade in PDAC promotes VTE, exacerbates tumor progression, and thus triggers tumor growth. The expanding tumor, in turn, spurs an increased likelihood of thromboembolic events [19]. A recent study found that symptomatic or asymptomatic VTE events are associated with significant mortality in PDAC patients and an overall worse prognosis. Only 20% of patients diagnosed with PDAC are eligible for surgical resection, the only treatment option that represents a potential cure [20]. Due to advances in neoadjuvant chemotherapy options, several patients diagnosed with local, resectable pancreatic tumors are treated with chemotherapy prior to surgery [21]. Patients treated with a combination of neoadjuvant therapy and surgical resection of pancreatic tumors have lower rates of VTE compared to patients with advanced, metastatic PDAC [22]. Thus, current guidelines suggest thromboprophylaxis for patients with advanced and metastatic PDAC who are treated with chemotherapy. There may be an association between more biologically aggressive tumors or those with progressive disease and a higher incidence of VTE [23]. Given the complexity of PDAC, addressing the status of coagulation in PDAC patients is complex in the current clinical setting.

## 2. Results

### 2.1. The Tumor Microenvironment and Hypercoagulability

The tumor microenvironment is an essential contributor to the hypercoagulable state of patients with cancer. Bidirectional crosstalk between tumor cells and a dynamic tumor microenvironment (TME) activates Virchow’s triad (coagulation activation, vessel damage, and stasis), which, in turn, promotes CAT or elevates VTE risk [17]. Virchow’s triad is still considered the gold standard for accessing VTE risk [24]. CAT and TME create a feedback loop and prime an extremely hostile TME in PDAC. The collaboration between TME, tumor cells, and different factors in the coagulation cascade contributes to the pathogenesis and associated clinical symptoms experienced by patients with VTE [25]. The advent of knockout mouse models further supports the bifunctional role of clotting proteins in mediating cancer growth and metastasis [17]. The prothrombotic properties of neoplastic tissues have an important pathogenic role. Several inflammatory mediators are associated with this prothrombotic state [17].

### 2.2. Altered Whole-Blood Parameters in PDAC Patients

#### 2.2.1. Red Blood Cells

In general, patients diagnosed with cancer who receive red blood cell transfusions are at an increased risk of VTE [26]. During storage at blood banks, RBCs undergo significant oxidative damage and loss of adenosine triphosphate [27]. These conditions facilitate the fraction of phosphatidyl serine (PhS)-exposing RBCs and the concentration of vesicles containing PhS. Given that PhS promotes coagulation, this could promote VTE in patients with cancer [28]. Uniquely, tumors in general often release PhS-containing micro vesicles, which also facilitate coagulation [29].

#### 2.2.2. Platelet Hyperactivity and Platelet Biomarkers

Hyperactivated platelets are also proposed to be a key process in promoting hypercoagulable state in PDAC patients. Several molecular pathways are involved in tumor cell induced platelet aggregation (TCIPA) in pancreatic cancer [30]. A recent study showed that patients with PDAC had a faster onset of whole-blood thrombin generation and overall increased endogenous thrombin potential when compared to plasma [31]. Interestingly, patients with PDAC are associated with hyperresponsive platelets, which degranulate in a shorter time. These parameters are associated with an increased risk of thromboembolic events [31]. Activated platelets secrete vascular endothelial growth factor (VEGF) and platelet-derived growth factor (PDGF). VEGF is an inducer of angiogenesis or new vessel formation, a mechanism adopted by tumor cells to promote growth and tumor neovascularization [32]. The platelet-derived growth factor (PDGF) signaling pathway plays a pivotal role in the progression of pancreatic cancer [33]. High levels of platelet activation markers are associated with an increased risk of venous thrombosis and shorter survival time in patients with PDAC. A recent study showed that platelet factor 4 (PF4) is a potential biomarker in PDAC. Higher levels of PF4 portend poor prognosis in PDAC patients and a higher likelihood of VTE development [34]. Platelets also store transforming growth factor-β (TGF-β) and secrete TGF-β into the circulation during platelet activation. TGF-β is one of the most potent inducers of epithelial to mesenchymal transition (EMT), one of the hallmarks of tumor metastasis [35]. PDAC tumor cells release microparticles containing podoplanin, but the association between this protein and VTE has not yet been determined [36]. Platelet aggregation-inducing factor, podoplanin, frequently upregulated on tumour cell surfaces, was found to not only trigger the formation of tumor–platelet crosstalk but also facilitate EMT by inducing TGF-β release from platelets [35]. Furthermore, P-selectin, a cell adhesion molecule, is a plasma biomarker for platelet and endothelial cell activation in coagulation. P-selectin, a member of the selectin family, is expressed on platelets and endothelial cells upon activation during the coagulation cascade and promotes platelet aggregation [37]. Similarly, elevated levels of P-selectin are associated with an increased risk of VTE and an increased risk of death in patients with PDAC [38].

Another essential protein dysregulation in PDAC patients is von Willebrand Factor (VWF). VWF is significantly elevated in PDAC patients, while levels of the enzyme that inactivates VWF, ADAMTS13, are lower [39]. In a study comprising 194 cancer patients diagnosed between January 2013 and September 2018, gastric, esophageal, colorectal, and pancreatic cancer had significantly higher VWF values compared with the control group [40]. Notably, the same study revealed a significantly inverse correlation (Spearman’s ρ = −0.41; *p* ≤ 0.001) between VWF values and ADAMTS-13 activity [40].

Furthermore, cancer-associated polyphosphate (Poly-P) secreted by cancer cells makes fibrin more resistant to fibrinolysis, strengthening clot formation [41]. The elevated expression of cell–cell adhesion molecules on tumor cells promotes interaction with endothelial cells, platelets, leukocytes, and erythrocytes, thereby facilitating clot localization [42].

#### 2.2.3. Coagulants and Anticoagulants

An analysis of the plasma of patients diagnosed with PDAC showed an increase in complexes containing kallikrein, Factor IX, Factor XIIa, Factor Xa, and their natural inhibitors. This suggests that the contact system and intrinsic pathway are activated in patients with pancreatic cancer [43]. Additionally, the release of pro-clotting proteins, contained in micro-vesicles or released without micro-vesicles, promotes thrombosis and inflammation [44]. These released micro-vesicles promote adhesion molecules between cells, allowing the direct interaction of tumor cells with several components involved in the clotting process. The release of these micro-vesicles fosters clotting activation locally and systemically in a malignant state [45]. Pancreatic cancer is often associated with elevated plasma levels of fibrinogen, factor (F) VIII, and D-dimers along with reduced levels of protein C and antithrombin III [4]. In general, tumor cells release interleukin-1 (IL-1), vascular endothelial growth factor (VEGF), tumor necrosis factor α(TNF-α), and other signaling molecules to induce inflammation in addition to coagulation [46].

A clotting protein of key importance in promoting coagulation as well as supporting tumor progression is Tissue Factor (TF) [47]. The release of TF from PDAC cells through micro-vesicles increases tumor aggressiveness by promoting favorable conditions for angiogenesis and metastasis. In particular, TF activates the extrinsic coagulation pathway and thus significantly elevates the risk of VTE in cancer patients [48]. Not only do elevated amounts of TF promote clot formation in mice, but TF is also a marker of cancer stem cells (CSCs) and may show relevance for the vascular generation and tumor resistance precipitated by these cells [49]. The seminal work of Simon Karpatkin established how thrombin binding to endothelium PAR-1 promotes coagulation changes in mouse tumor models [50]. This exciting breakthrough further established that the activation of the coagulation cascade and platelets are early events in oncogenesis, create a conducive microenvironment for the development of cancer, and are susceptible to regulation by oncogenes [17].

Notably, pancreatic cancer cells have been shown to secrete mucins, high-molecular-weight glycoproteins, into the circulation, and elevated levels of mucins are indicative of cancer progression. Carcinoma mucins trigger microthrombi formation in murine models [51]. A recent study elucidated the role of a natural anticoagulant, Protein S (PS), in PDAC [52]. PS is downregulated during the progression of PDAC [53]. PS is a protein synthesized in the liver and is essential for hemostasis control [54]. Additionally, PS is a signaling molecule that regulates growth by binding to a group of tyrosine kinase receptors, collectively known as TAM (Tyr, Axl, Mer) receptors [55]. In the circulation, about 60% of total protein S is bound to C4b-binding protein (C4BP) [56], an acute phase complement component. Because only 40% of the protein S that is free is functionally active, only patients with low free protein S levels are prone to venous thrombosis. Therefore, the diagnosis of protein S deficiency requires the measurement of both free and bound forms of protein S [56]. In several cancers, including PDAC, TAM receptors are upregulated due to their function in promoting cell survival, proliferation, efferocytosis, and apoptosis [57]. However, targeting TAM receptors in anti-cancer treatments has had limited success due to the multifunctional roles of TAM receptors. PS-TAM interaction, primarily occurring through the Mer receptor of the TAM family, promotes apoptosis in pancreatic cancer cell lines [53]. Pillai et al. [52] showed the efficacy of PS as a potential agent to limit PDAC progression in PC cell lines. Notably, PDAC is commonly characterized by numerous and severe hypoxic regions, with a median tissue partial oxygen pressure (pO2) of 0–5.3 mmHg (0–0.7%) compared to the adjacent normal pancreas pO2 at 24.3–92.7 mmHg (3.2–12.3%) [58]. One hallmark of PDAC tumors is the presence of desmoplastic fibrotic stroma that creates a physical and immunological barrier and promotes hypoxia and tumor progression [2]. Hypoxia is considered as one of the independent prognostic factors for PC [59]. Recent evidence supports a drop in the anticoagulant PS level in obese murine livers [52]. Hypoxia elevates thrombotic risk [52]. This recent study provides a potential opportunity to target both hypoxia and the associated risk of thromboembolism in PDAC by PS supplementation.

#### 2.2.4. Neutrophil Extracellular Traps (NETs)

Surprisingly, elevated white blood cell (WBC) counts, also known as leukocytosis, are associated with an increased incidence of VTE in patients with cancer in general. Patients with leukocytosis after the first cycle of chemotherapy had a significantly higher risk of VTE when compared to cancer patients with normal leukocyte counts. In PDAC patients, the risk of recurrent VTE is increased by 1.6-fold, triggered by leukocytosis [60]. The formation of neutrophil extracellular traps (NETs) is one underlying mechanism by which this phenomenon occurs, which also promotes thrombosis. Neutrophils are the most common WBC and play a critical role in the immune response against microbes. Activated neutrophils from NETs induce inflammation through reactive oxygen species production and associated platelet activation [61]. Notably, pancreatic cell lines are known to stimulate the formation of NETs [62]. Radiation and other targeted therapies can also disrupt vascular endothelium, further contributing to NET formation [63]. Notably, chemotherapy damages the endothelial cell surface, thus releasing neutrophil extracellular traps (NETs) [64]. Oto et al. [65] validated a cluster of seven microRNAs as predictive markers of venous thrombosis in PDAC patients. This is the reported study that established the potential of plasma miRNAs and neutrophil activation markers to predict VTE in PDAC.

### 2.3. Hypercoagulability Risk Factors and Risk Assessment

In general, there are several patient characteristics which may elevate their risk of VTE [66]. Some risk factors include age greater than 65 years old and immobilization [44]. Thrombosis risk increases with age, a trend which is also seen in patients with cancer. Immobilization due to hospitalization or other factors also poses a significant risk for the occurrence of VTE. The prevalence of VTE incidence in hospitalized patients with pancreatic cancer is 8% and is associated with significant in-hospital mortality. In cancer patients with poor mobility and bed rest for more than three days, VTE rates are significantly elevated [67]. Notably, individuals of Black ethnicity are at an increased risk of VTE occurrence [68].

Smoking is also an independent risk factor for developing PDAC, as with other cancers, and with elevated VTE risk [69]. Female sex also has a deleterious effect for patients diagnosed with PDAC and in other cancer types as well [68]. A current diagnosis of cancer with another comorbidity significantly raises the risk of VTE [70]. Obesity, pulmonary disease, renal disease, infection, and anemia are comorbidities commonly associated with VTE in patients with cancer [71]. Additionally, chemotherapy and hormonal therapy to treat cancer are associated with elevated thrombotic risk [72]. Overall, cancer patients with a history of VTE have a 6-fold higher risk of developing VTE [73]. Particularly in pancreatic cancer patients, the risk of VTE elevates with identified stage IV disease, planned surgery, and unresectable disease [74].

Malignancy is often associated with several hemostatic abnormalities seen in laboratory screening. While these abnormalities vary among cancer patients, abnormalities will worsen with cancer progression and the risk of hypercoagulability increases with elevated tumor burden [75]. One example of a laboratory abnormality which is common in several tumors is the release of TF [76]. The release of cellular fragments or platelets is increased in highly severe or metastatic disease, promoting coagulation locally and systemically. The release of TF, which is also found on activated platelets or endothelial cells, increases the risk of VTE by 2.5 times. Not only does TF activate the coagulation cascade, but TF also enables immune avoidance by the tumor cells and promotes disease progression as a result [77]. Limited evidence suggests that a low level of tissue factor pathway inhibitor (TFPI), a potent inhibitor of TF, elevates the risk of VTE [78]. However, these results were not reproduced in a longitudinal study [78]. TFPI was independently associated with risk of VTE and all-cause mortality in patients with cancer. In a prospective observational cohort study with the primary outcome VTE, Englisch et al. [79] observed that tumor types known to have a high risk of VTE, such as gastroesophageal, pancreatic, and brain cancers, were associated with higher levels of TFPI. This sounds counterintuitive, given TFPI is an anticoagulant. However, Englisch et al. [79] reasoned that, in these patients, the TFPI does not primarily confine in the endothelium and higher levels of TFPI were detected in circulation during cancer progression. Indeed, higher levels of circulating TFPI were observed in patients with metastatic disease compared to patients with non-metastatic disease in the same study [79]. Furthermore, the abnormal coagulation parameters associated with PDAC have also brought to attention to the potential of using assorted coagulation parameters, in addition to US FDA approved serum cancer antigen CA19-9 marker, as a viable screening method for PDAC [4]. An elevated CA19-9 level was a significant risk factor for VTE and the occurrence of VTE was, overall, associated with poor prognosis in PDAC patients [80]. Another study showed that CA19-9 levels increase with the extent of VTE and are higher in PDAC patients with VTE compared to PDAC patients without VTE [81]. In patients being considered for the surgical resection of a pancreatic tumor, a preoperative high serum D-dimer level over 1.0 ug/mL is associated with metastasis and is an unfavorable prognostic marker of overall survival in PDAC patients [82]. Likewise, patients with high platelet counts have a higher rate of VTE when compared to patients with normal platelet counts [83]. Patients with elevated PF4 prior to chemoradiation treatment have a 2.7-fold increased risk of VTE development [34]. The incidence of VTE with pancreatic cancer is also associated with tumor location. Tumors in the tail (cauda) and body (corpus) of the pancreas are associated with elevated VTE risk compared to tumors in the head (caput) of the pancreas [13]. Undergoing cancer treatment also drives VTE risk higher in any cancer type [84]. Patients with resectable PDAC are given FOLFIRINOX chemotherapy or gemcitabine-based chemoradiation as first line treatment. In cancers overall, gemcitabine, 5-FU, and oxaliplatin are associated with thrombosis [85].

Additionally, patients with an elevated peak thrombin generation level (≥611 nM) in plasma have an increased rate of VTE [86]. Notably, Plasminogen Activator Inhibitor 1 (PAI-1) can be released by pancreatic tumor cells, as well as by activated platelets, to inhibit fibrinolysis or clot retraction [6]. High levels of PAI-1 can inhibit plasminogen activation, thereby exacerbating the expansion and persistence of clots in PDAC [6].

### 2.4. Risk Assessment of VTE

VTE during chemotherapy was associated with a 2.5-fold decrease in progression-free survival (PFS) and a 1.6-fold risk decrease in overall survival (OS) in a retrospective cohort of 227 patients with unresectable PC, as reported by Frere [87]. Similarly, in a small cohort of 135 PC patients, the onset of VTE was significantly associated with increased mortality [71]. To investigate the incidence and risk factors for the onset of VTE regardless of the stage or treatment, and to further assess their clinical outcomes, a prospective, multicenter BACAP-VTE (Base Clinico-Biologique de l’Adénocarcinome Pancréatique-Venous Thromboembolism) study was conducted on 731 newly diagnosed PDAC patients from May 2014 to November 2018 in France [13]. The primary endpoint was the onset of VTE during follow-up. The secondary endpoints were progression-free survival (PFS) and overall survival (OS) times. The study found that 152 patients (20.79%) developed a VTE during a median follow-up of 19.3 months. The study further documented that the median time from PDAC diagnosis to the onset of VTE was 4.49 months. The cumulative incidence values of VTE were 8.07% (95% confidence interval [CI], 6.31–10.29) at 3 months and 19.21% (95% CI, 16.27–22.62) at 12 months. The study illustrated that frequent and early onsets of VTE after diagnoses of PDAC are associated with significant decreases in times of PFS and OS [13]. Notably, in the BACAP-VTE study, approximately 50% of the VTE events were incidentally diagnosed. This finding is consistent with previous studies, where incidental VTE accounted for 30–50% of VTE events among patients with PDAC [13,88,89]. This reported incidence of VTE in the BACAP-VTE study is higher than that in the PRODIGE 4/ACCORD 11 and PRODIGE 24/ACCORD 24 randomized controlled trials, which showed the survival benefits from adjuvant chemotherapy with FOLFIRINOX compared to gemcitabine (GEM) alone in metastatic and in resected PDAC patients, respectively, because these later studies accounted for symptomatic VTE only [13].

Many risk assessment models (RAMs) have been developed over the last decade to assess the risk of VTE in cancer patients. However, none of these RAM was designed to specifically assess this risk in PC patients. The Caprini score is the most widely used RAM for assessing the risk of VTE in patients undergoing surgery. It has been validated in several types of cancers [65]. However, this model failed to identify patients at the highest risk of VTE in a retrospective cohort of 426 PC patients undergoing preoperative treatment followed by surgical resection [66]. However, the Khorana score is the most validated tool. The Khorana score is a predictive clinical score developed to stratify ambulatory cancer patients at high risk of venous thromboembolism (VTE), who may be eligible for thromboprophylaxis [89]. At baseline, the Khorana score classified patients as ‘intermediate risk’ (2 points) or ‘high risk’ (≥3 points) for VTE. This risk assessment is necessary to determine if a patient follows the clinical guidelines to be eligible for pharmacological thromboprophylaxis with either a reduced dose direct oral anticoagulants (DOACs) or a prophylactic dose of low-molecular-weight heparin (LMWH) to prevent VTE. Ambulatory cancer patients are judged to be at high risk of VTE. The Khorana score uses a positivity threshold of two points. At two points, patients with pancreatic cancer are considered at high risk for VTE and eligible for thromboprophylaxis [90]. The Khorana score factors in significant changes in cellular biomarkers associated with PDAC-related venous thrombosis. For example, anemia with a hemoglobin level below 6.2 mmol/L is used in the Khorana score as a risk score for VTE development in all cancer patients. The Khorana score includes a platelet count of ≥350 × 10^9^/L as an independent risk factor for VTE [91]. In the multicenter BACAP-VTE study [13], the Khorana risk assessment score did not discriminate between patients with intermediate vs. high VTE risk scores; a total of 21% of patients with a score of 2, and 18% of those with a score of 3 or higher, developed VTE. These results are consistent with those of previous small retrospective studies [13,89]. The results might partly be explained by the lack of predictive value of several other items, such as BMI and hemoglobin level, in PDAC patients which were not associated with the onset of VTE by multivariate analysis in the BACAP-VTE study [13]. Furthermore, the Khorana score was developed to predict the VTE risk in patients with cancers in general, and only 2% of all the patients in the original cohort had pancreatic cancer [92]. In patients with PDAC undergoing chemotherapy, no difference was found in the rates of VTE between intermediate- and high-risk patients, as estimated by the Khorana score.

In a clinical setting, gemcitabine (GEM) plus nab-paclitaxel (GnP) combination chemotherapy is a standard regimen for unresectable pancreatic cancer patients with a good performance status (PS) [93]. In the phase 3 MPACT trial, out of the 174 patients, 17 patients (9.8%) were classified in the VTE (+) group, and 3 of them were symptomatic. Thirteen patients were diagnosed with VTE at treatment initiation. Four patients were diagnosed after treatment initiation, and the median time to diagnosis after treatment initiation was 55 days [94]. This study reveals that the early detection of VTE was associated with a poor prognosis in patients with unresectable metastatic pancreatic cancer (UR-MPC) who underwent GnP as a first-line chemotherapy. Mandala et al. [95] also reported that the presence of synchronous VTE at cancer diagnosis was associated with a higher probability of treatment unresponsiveness (odds ratio [OR] 2.98, 95% CI 1.42–6.27, *p* = 0.004), but were not associated with PFS or OS upon multivariate analysis. The same study [95] further revealed that the occurrence of a VTE during chemotherapy was associated with significantly shorter PFS (HR 2.59, 95% CI 1.69–3.97, *p* < 0.0001) and OS (HR 1.64, 95% CI 1.04–2.58, *p* = 0.032). Although VTE is considered a critical complication in patients with UR-MPC, the impact of early VTE detection remains to be fully elucidated in patients who receive multidrug combination therapy, including GnP. Yamai et al. [94] observed that the OS and PFS were significantly shorter in the VTE (+) group in this study. The OS and PFS of the VTE (−) and VTE (+) groups were 400 vs. 259 days (OS; *p* < 0.001) and 162 vs. 120 days (PFS; *p* = 0.008), respectively. In a Japanese phase 1/2 study, Ueno et al. [96] observed how GnP chemotherapy led to median PFS and OS times of 6.5 months (95% CI, 5.1–8.3) and 13.5 months (95% CI, 10.6—not reached), respectively. Yamai et al.’s study [94] results showed that the OS and PFS in the VTE (−) group were consistent with those in the Japanese phase 1/2 study, and the OS and PFS were both significantly shorter in the VTE (+) group than in the VTE (−) group.

### 2.5. Prophylaxis and Treatment of VTE in PDAC Patients

#### 2.5.1. Thromboprophylaxis

The use of primary thromboprophylaxis, a supportive treatment with potentially significant clinical benefit, continues to be underrecognized in treating pancreatic cancer patients [6]. Most anticoagulants used clinically aim to reduce thrombin generation. Thrombin has over a dozen recognized substrates and has variable effects on cancer progression. Several studies evaluated the effectiveness of the targeted inhibition of the coagulation cascade to improve survival in cancer patients. However, few data were obtained from PC patients due to their short life expectancy [6]. Since 2013, the international initiative on thrombosis and cancer (ITAC) clinical practice guidelines have recommended the use of thromboprophylaxis with LMWH in surgical PC patients undergoing major surgery, hospitalized patients with acute medical illness and reduced mobility [97,98], and in locally advanced or metastatic ambulatory PC patients receiving chemotherapy [97,98]. Four authoritative clinical practice guidelines (CPGs), including the International Initiative on Thrombosis and Cancer (ITAC), the American Society of Clinical Oncology (ASCO), the American Society of Hematology (ASH), and the National Comprehensive Cancer Network (NCCN), routinely meet and recommend the clinical practices to pursue in different clinical settings for cancer-associated thrombosis [99]. The most recent review by Frere et al. [99] details the consensus and divergence of the guidelines issued by the respective bodies. Routine thromboprophylaxis for unselected ambulatory cancer patients with LMWH is not recommended by major guidelines. However, in selected high-risk patients, such as patients with advanced pancreatic cancer starting systemic therapy, ISTH suggested higher doses of LMWH as outpatient thromboprophylaxis in the 2014 guidance document [100]. The bleeding risk associated with standard anticoagulants significantly limits their concomitant use with other anticancer agents [100]. Patients undergoing surgical resection of pancreatic tumors require thromboprophylaxis with low-molecular-weight heparin (LWMH). For individuals hospitalized due to malignancy or who are undergoing systemic anticancer treatment, factor Xa inhibitors, apixaban or rivaroxaban are recommended [6]. However, these patients must be risk-stratified to determine correct dosages [101]. There are two broad classes of direct oral anticoagulants (DOACs); one class aims to limit thrombin generation by inhibiting FXa and the other class directly inhibits thrombin proteolytic activity. The net clinical benefit of primary thromboprophylaxis in advanced PC patients has been firmly established in two pivotal randomized control trials [102], which specifically addressed the efficacy and safety of LMWH in this setting [103]. The FRAGEM trial randomized 123 advanced PC patients to receive gemcitabine plus weight-adjusted therapeutic doses of dalteparin, a LMWH, for 12 weeks or gemcitabine alone [102]. The co-primary endpoints were the rate of symptomatic or incidentally diagnosed VTE events during the 12-week LMWH period and the rate of symptomatic or incidentally diagnosed VTE events during the overall follow-up period. The rate of VTE was significantly lower in the dalteparin arm (3.4% vs. 23% in the control arm, risk ratio 0.145, 95% CI: 0.035–0.612, *p* = 0.002) [102]. The PROSPECT-CONKO 004 trial randomized 312 advanced PC patients to receive supra-prophylactic doses of enoxaparin, another LMWH, during the first 3 months of chemotherapy or chemotherapy alone [103]. Unlike in FRAGEM, incidental VTE events were excluded from the analysis in this PROSPECT-CONKO 004 trial. The cumulative incidence rate of symptomatic VTE within the first 3 months was 1.3% in the enoxaparin arm compared to 10.2% in the control arm (HR 0.12, 95% CI: 0.03–0.52). The rates of major bleeding events were similar in both arms. The PFS and OS did not differ between the two arms [103]. Notably, two high-grade randomized placebo-controlled trials assessed the efficacy and safety of primary thromboprophylaxis with prophylactic doses of 2.5 mg twice daily for up to 6 months of apixaban, a factor Xa inhibitor in the AVERT trial [104], and 10 mg once daily for up to 6 months of rivaroxaban, another factor Xa inhibitor in the CASSINI trial [105], in cancer patients with a Khorana score ≥ 2 undergoing chemotherapy. The results from a subgroup of PC patients were reported only for the CASSINI trial [106]. Among the 273 PC patients included in this prespecified subgroup analysis, 214 (78%) had a locally advanced or metastatic PC and 271 (99.3%) were receiving cytotoxic chemotherapy (fluorouracil-based in 47.6% of cases and gemcitabine-based in 44.7% of cases). Rivaroxaban did not significantly reduce the rates of the primary efficacy endpoint of symptomatic DVT, asymptomatic proximal DVT, or any PE- and VTE-related death within the 6-month observation period (absolute difference of 3.4%, *p* = not significant). However, most of the VTE events occurred after the discontinuation of rivaroxaban (61.5%) compared to the placebo (22.2%). The newer direct thrombin inhibitor or factor Xa inhibitors carry less bleeding risk than older agents such as heparins and warfarin, but there are limited data regarding the use of these agents in cancer patients [107,108]. However, overall primary thromboprophylaxis in ambulatory cancer patients treated with chemotherapy, and in PC patients with an elevated VTE risk, is safe, effective, and advisable [22]. With regard to cancer patients, there are no relevant data for dabigatran in the primary prevention or treatment of VTE. For the primary prevention of VTE in cancer patients, evidence is only available for direct factor Xa inhibitors, apixaban and rivaroxaban, in a phase III study [100].

#### 2.5.2. Treatment of Established VTE

Early retrospective studies reported no association between VTE and overall survival (OS) in PC patients [14,109]. However, all patients included in these studies had metastatic disease with a short life expectancy. By contrast, later high-quality studies reported that the onset of VTE was associated with a poorer prognosis. In a retrospective cohort of 227 patients with unresectable PC, VTE during the course of chemotherapy was associated with a 2.5-fold decrease in progression-free survival (PFS) and a 1.6-fold risk decrease in OS [95]. The potential for anticoagulants to limit cancer progression was not shown in all studies, and whatever benefit has been observed appears to be dependent on multiple factors, including the type of cancer, stage, and the specific anticoagulant used for therapy [19]. Yamai et al.’s study [94] mentioned how all the patients in the VTE (+) group were treated with a direct oral anticoagulant (DOAC) or unfractionated heparin from the time of diagnosis, and all the patients continued to receive thromboprophylaxis unless they experienced severe anemia. Notably, in a small cohort of 135 PC patients, VTE was associated with increased mortality and anticoagulant therapy improved survival in those patients with VTE (HR 0.30, 95% CI 0.12–0.74, *p* = 0.009) [71]. The ability to use oral-only anticoagulation strategies, forgoing the need for long-term daily injection and dose adjustment, may seem appealing, but adds to the complexity of decision making. The appropriate selection of anticoagulants is imperative in implementing high-quality care for cancer patients with VTE [87,110]. The treatment of cancer-associated VTE remains a challenge. In these patients, personalized therapy is recommended. Personalized therapy based on the type and stage of cancer, cancer treatment, potential drug–drug interactions, bleeding risk, and the patient’s preference should be considered [111].

The ITAC CPGs recommend using LMWH for the initial and long-term treatment of established VTE when creatinine clearance is ≥ 30 mL per min (Grade 1B) [87]. For patients without risk of gastrointestinal or genitourinary bleeding, rivaroxaban (in the first 10 d) or edoxaban (started after at least 5 day of parenteral anticoagulation) can also be used (Grade 1B) [87]. Unfractionated heparin (UFH) provides an alternative option when LMWH or DOACs are contraindicated or not available (Grade 2C) [87]. Anticoagulation should be continued for at least 6 months (Grade 1A) or indefinitely while cancer is active or treated [8]. There are sufficient data for the treatment of cancer-associated VTE with direct factor Xa inhibitors—apixaban, edoxaban, and rivaroxaban [112]. Briefly, the direct factor Xa inhibitors are a reasonable option in ambulatory PC patients with DVT or PE with an intact upper gastrointestinal tract, without nausea or vomiting, with a low risk of bleeding, with a platelet count > 50,000/mm^3^, with a creatinine clearance > 30 mL/min, without severe hepatic impairment, and for whom no surgical intervention was planned. They should not be used in patients with creatinine clearance < 30 mL/min, luminal gastrointestinal lesion, platelet count < 50,000/mm^3^, high bleeding risk, recent or planned surgery, or potential drug–drug interactions [87]. The coadministration of any of the factor Xa inhibitors or direct thrombin inhibitors is not recommended with cancer therapies that are strong P-glycoprotein inducers or inhibitors, as these DOACs, as a class, are a substrate for p-glycoprotein [113]. DOACs are also subject to variable metabolism via the cytochrome P450 (CYP) system. Several cancer and adjunctive therapies (including anti-emetics, opioids, and antibiotics) can alter CYP3A4 metabolism [113] and, therefore, careful consideration is encouraged before administering these therapies together with DOACs. The International Initiative on Thrombosis and Cancer (ITAC) is an independent academic working group of experts aimed at establishing a global consensus for the treatment and prophylaxis of cancer-associated thrombosis. The 2019 ITAC guidelines for treating cancer-associated VTE were succinctly documented by Farge et al. [114]. The COVID-19 pandemic added a layer of complexity in the most recent 2022 ITAC guidelines and, therefore, it is recommended to adhere to the 2019 guidelines in a post pandemic world.

#### 2.5.3. Potential Effects of Anticoagulation Therapy on Cancer Progression

The bleeding risk and the quality of life due to LMWH subcutaneous injections are two major concerns in patients with PDAC, although Farge et al. [115] showed that quality of life does improve in patients with cancer receiving LMWH for the treatment of a VTE event. In a prespecified subgroup in the CASSINI trial, as mentioned above, the appropriate use of primary thromboprophylaxis effectively reduced the rate of VTE onset in patients with PDAC. These encouraging results led the International Initiative on Thrombosis and Cancer [114] and American Society of Clinical Oncology Clinical Practice [116] to recommend anticoagulation therapy with apixaban or rivaroxaban in cancer outpatients undergoing chemotherapy with a Khorana score ≥ 2, no bleeding risk, and no drug–drug interactions (Grade 1B) [114]. Thus, these guidelines indicate that thromboprophylaxis may now be considered in all ambulatory pancreatic cancer patients, given that two points are assigned for the primary site being the pancreas in the Khorana score [92].

A recent meta-analysis of several clinical trials suggested that DOACs are effective at preventing the occurrence of VT [117]. It was shown that treating patients who have an active cancer diagnosis with apixaban, a Factor Xa inhibitor, is associated with a reduced risk of recurrent VTE without increasing bleeding risk, compared to other DOACs such as edoxaban and rivaroxaban [117]. The metanalysis was conducted over several studies, studying the efficacy among several cancer types, with the primary safety outcome as major bleeding [117]. Overall, current guidelines suggest the use of apixaban, rivaroxaban, and LMWH in patients without critical risk factors for bleeding and without the potential for drug–drug interactions [22]. Another treatment option in thromboprophylaxis is warfarin, a vitamin K antagonist that targets a broad array of vitamin-k-dependent coagulation proteins, including Factor IIa, Factor VIIa, Factor IXa, and Factor Xa. However, pleiotropic target engagement, elevated bleeding risk, significant adverse events, and the close monitoring of INR are associated with warfarin. Therefore, current guidelines recommend the use of DOACs instead. All these treatments, however, are associated with increased bleeding risk [118]. There is no clear evidence regarding whether thromboprophylaxis improves survival in at-risk PDAC patients. Some studies claim there is no evidence that thromboprophylaxis administration prolongs survival in patients with advanced PDAC and chemotherapy [22]. However, some other recent studies have shown improved overall survival in patients receiving warfarin compared to LMWH long term [119]. There is an urgent medical need to determine the ultimate long-term clinical utility of prophylactic anticoagulants in individuals diagnosed with PDAC. Figure 1 depicts the various factors that increase VTE risk.

## 3. Discussion

Increased procoagulant activity, decreased anticoagulant activity, and impaired fibrinolysis (resolution of clots) are key contributors in PDAC progression. The association between PDAC and venous thromboembolism is a significant cause of mortality. Several components of the tumor microenvironment collaborate to promote a hypercoagulable environment, which causes the clinical syndromes so often experienced by patients diagnosed with PDAC. Hemodynamic alterations of whole blood parameters, platelet hyperactivity, and coagulant and anticoagulant proteins promote local and systemic clotting activation. There are several individual and genetic risk factors, as well as risk factors posed in the process of treating PDAC, that promote VTE. Clinically, the Khorana risk score is used to assess the use of thromboprophylaxis in treating PDAC patients with or without surgically resectable tumors. The use of DOACs and LWMH prophylactically is recommended to prevent morbidity and mortality from VTE in candidate patients diagnosed with PDAC. Ultimately, the interplay of VTE and PDAC poses a complex clinical and physiological dilemma which necessitates further insight and warrants a precision medicine approach in clinical settings. The summary of the review is shown in Figure 2.

## Figures and Tables

**Figure 1 ijms-25-05661-f001:**
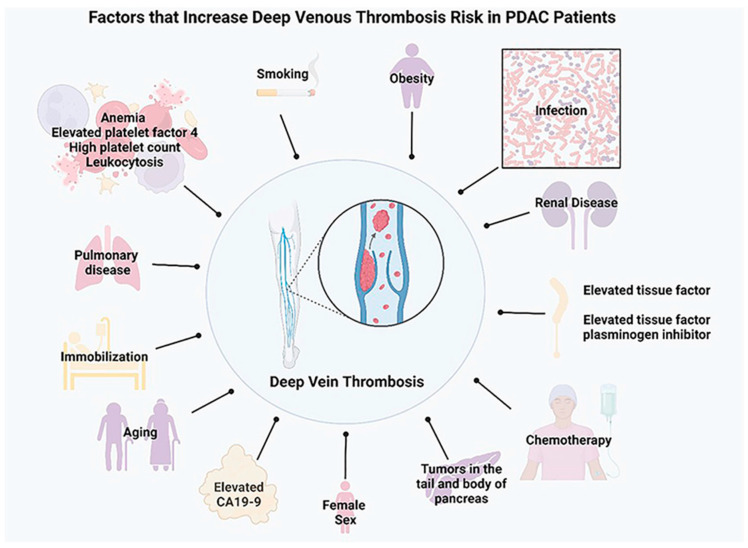
Factors that increase VTE risk in PDAC patients (created using BioRender.com).

**Figure 2 ijms-25-05661-f002:**
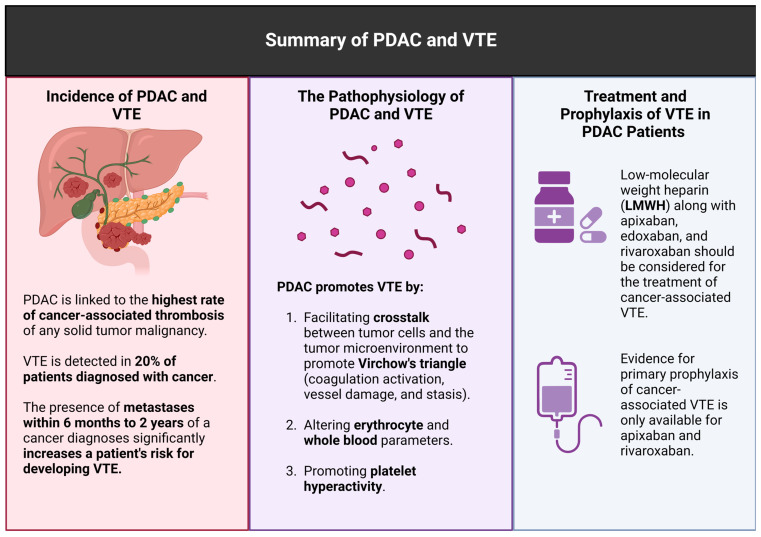
Summary of PDAC and VTE (created using BioRender.com).

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
