# Peer review of "Pancreatic Cancer and Venous Thromboembolism"

_ijms, 2024, doi:10.3390/ijms25115661_

Round 1

Reviewer 1 Report

Comments and Suggestions for Authors

Content suggestions:

1.         I would like to kindly ask the Authors to add the details about the correlation of the onset and treatment of VTE in the patients with pancreatic cancer treated with chemotherapy (after the chemotherapy, there is thrombocytopenia...). Does administration of chemotherapy influence the prothrombotic risk ?

2.         Which drugs are preferred in the management of pulmonary embolism or VTE in general in these patients ? LMWH or DOACs ?

Author Response

We appreciate the Reviewer #1 feedback and constructive criticism. 

"I would like to kindly ask the Authors to add the details about the correlation of the onset and treatment of VTE in the patients with pancreatic cancer treated with chemotherapy (after the chemotherapy, there is thrombocytopenia...). Does administration of chemotherapy influence the prothrombotic risk"- As per Reviewers#1 suggestion, we incorporated relevant information under 2.7. Risk Assessment of VTE  

"Which drugs are preferred in the management of pulmonary embolism or VTE in general in these patients ? LMWH or DOACs ?"-As per Reviewer #1 suggestion , we incorporated relevant information and literature under 2.8. Prophylaxis and Treatment of VTE in PDAC Patients.

We sincerely thank Reviewer #1 for these valuable feedback to strengthen the manuscript. 

Reviewer 2 Report

Comments and Suggestions for Authors

The authors provide an interesting paper on "pancreatic cancer (PaCa) and  thromboembolism".  This title may be misinterpreted by potential readers because it does not (i) deal with arterial thromboembolism (elevated risk in PaCa), (ii) give more than a few words to superficial venous thrombosis (a clinically relevant problem in PaCa) and (iii) give any advice how to treat PaCa-associated vte. Thus "pancreatic cancer and  venous thromboembolism - pathophysiology and primary prevention" or so may be more apropriate.

As there are different direct oral anticoagulant drugs" (DOAC) available on the market and others may become available in the near future, I strongly recommend to use clear and correct designations such as factor Xa-inhibitor or the specific drugs used in the trials (no higher grade data on dabigatran, a thrombin inhibitor, in PaCa, or on edoxaban for primary prevention in PaCa).

A reference should be given for the statement in line 54 / 55 .R

"Red blood cells (RBCs) are crucial in pathophysiology of cancer-associated thrombosis (CAT) by providing surface area necessary to facilitate the coagulation cascade [25]." I clearly disagree! The referenced paper is rather old and does not provide higher grade evidence for a "crucial" role. To my knowledge there are relevant clinical data on an increased vte-risk related to RBC only in anemia and after blood transfusion. Reference [26] is up to date, and points to an important role of platelets and not of RBC . Thus the text of 2.2. (line 82 ff) needs a thorough rewording.

In fact the following paragraphs (2.3. - 2.5) are subchapters of 2.2.!

lines 137 ff:

I miss some words on von willebrand factor. With regard to protein S, it should be clarified if "total" or "free" protein S is meant and mention the role of acut phase and C4-binding protein.

lines 205 ff:

age is a risk factor, but the risk does not change on the 65th birthday !

RAM are important, but - to my knowledge  - none has been  positively evaluated to differentiate vte-risk in PaCa. Otherwise please give the facts.

That PaCa is associated with 2 points in th Khorana score, resulting in increased vte risk should be mentioned when the KS is given for the first time,

Lines 270 ff

Overall paragraphs 2.6. to 2.8 do not clearly differentiate between what is known and recommended for "overall cancer patients" and what is evidence- based for PaCa, the only solid cancer where we have high grade evidence with regard to vte-prophylaxis !

There are important trial results on incidental deep vein thromboses and on pulmonary embolism in PaCa! Furthermore on primary vte-prevention in PaC with different dosages of LMWH and with rivaroxaban (less clear with apixaban!). (It may well be considered not to give these clinical data, but then the title should be limited to pathophysiology and paragraphs 2.6-2.8 should be minimized !)

"Clinically, the Khorana risk score is used to assess the use of thromboprophylaxis in treating PDAC patients with or without surgically resectable tumors". Totally wrong ! As mentioned by the authors patients with PaCa are at high risk to VTE . the authors then correctly: "The use of DOACs (???) and (or) LWMH prophylactically is recommended to prevent morbidity and mortality from VTE in candidate patients diagnosed with PDAC" Thus, there is no need to use the Khorana Score in PaCa (it is very useful for several other solid cancers!)

Author Response

We appreciate the constructive criticism and important suggestions by both the reviewers to strengthen the manuscript. Below is our point by point response to reviewer’s concerns,

Reviewer#2

 The authors provide an interesting paper on "pancreatic cancer (PaCa) and thromboembolism".  This title may be misinterpreted by potential readers because it does not (i) deal with arterial thromboembolism (elevated risk in PaCa), (ii) give more than a few words to superficial venous thrombosis (a clinically relevant problem in PaCa) and (iii) give any advice how to treat PaCa-associated vte. Thus "pancreatic cancer and venous thromboembolism - pathophysiology and primary prevention" or so may be more appropriate.

Ans: We totally agree. Accordingly, we modified the title of the manuscript.

As there are different direct oral anticoagulant drugs" (DOAC) available on the market and others may become available in the near future, I strongly recommend to use clear and correct designations such as factor Xa-inhibitor or the specific drugs used in the trials (no higher grade data on dabigatran, a thrombin inhibitor, in PaCa, or on edoxaban for primary prevention in PaCa).

Ans: The point is well taken, and we incorporated in the text wherever applicable.

A reference should be given for the statement in line 54 / 55. R

Ans: Done

"Red blood cells (RBCs) are crucial in pathophysiology of cancer-associated thrombosis (CAT) by providing surface area necessary to facilitate the coagulation cascade [25]." I clearly disagree! The referenced paper is rather old and does not provide higher grade evidence for a "crucial" role. To my knowledge there are relevant clinical data on an increased vte-risk related to RBC only in anemia and after blood transfusion. Reference [26] is up to date, and points to an important role of platelets and not of RBC . Thus the text of 2.2. (line 82 ff) needs a thorough rewording.

Ans: Thanks for clarifying this important aspect. We have incorporated reviewer’s suggestion and modified the text accordingly.

In fact the following paragraphs (2.3. - 2.5) are subchapters of 2.2.!

Ans: Done as per suggestion.

I miss some words on von willebrand factor. With regard to protein S, it should be clarified if "total" or "free" protein S is meant and mention the role of acut phase and C4-binding protein.

Ans: We explained and incorporated a few lines with reference as per your suggestion.

age is a risk factor, but the risk does not change on the 65th birthday !

Ans: We didn’t say that 65th year is an inflection point for PaCa incidence. We stated “Some risk factors include age greater than 65 years old”.

RAM are important, but - to my knowledge - none has been positively evaluated to differentiate vte-risk in PaCa. Otherwise please give the facts.

Ans: We tempered the wordings as per reviewer’s feedback.

That PaCa is associated with 2 points in th Khorana score, resulting in increased vte risk should be mentioned when the KS is given for the first time,

Ans: Incorporated accordingly

Overall paragraphs 2.6. to 2.8 do not clearly differentiate between what is known and recommended for "overall cancer patients" and what is evidence- based for PaCa, the only solid cancer where we have high grade evidence with regard to vte-prophylaxis!

Ans: We appreciate your suggestion and accordingly we modified the mentioned paragraphs (2.6 to 2.8)

There are important trial results on incidental deep vein thromboses and on pulmonary embolism in PaCa! Furthermore, on primary vte-prevention in PaC with different dosages of LMWH and with rivaroxaban (less clear with apixaban!). (It may well be considered not to give these clinical data, but then the title should be limited to pathophysiology and paragraphs 2.6-2.8 should be minimized !)

Ans: We deliberated and agreed to provide relevant literature rather than minimizing the paragraphs 2.6 to 2.8. We felt that any effort to minimize the paragraphs would minimize the relevance of the review. Therefore, we kept the full title as suggested by the reviewer at the beginning and added more literature on VTE prevention in PaCa.

"Clinically, the Khorana risk score is used to assess the use of thromboprophylaxis in treating PDAC patients with or without surgically resectable tumors". Totally wrong! As mentioned by the authors patients with PaCa are at high risk to VTE. the authors then correctly: "The use of DOACs (???) and (or) LWMH prophylactically is recommended to prevent morbidity and mortality from VTE in candidate patients diagnosed with PDAC" Thus, there is no need to use the Khorana Score in PaCa (it is very useful for several other solid cancers!)

Ans: We redacted the first line in our revised manuscript

Reviewer #3

line 49: ... such as the JAK-2V617F mutation in myeloproliferative neoplasms (MPN) ...

Ans: We agree, it is irrelevant and therefore we deleted it from our revised manuscript.

Page 2, line 52: What does "exacerbates tumor milieu" mean? This phrase is not very precise.

Ans: We changed “exacerbates tumor milieu” to “exacerbates tumor progression” in revised manuscript.”

Page 3, first paragraph: "In PDAC, tumor cell-induced platelet aggregation (TCIPA) can promote platelet aggregation via activation of thrombin." This sentence does not make any sense.

Ans: We made the necessary changes in the revised manuscript.

Page 3, first paragraph: "VWF is critical in promoting the clotting cascade and is significantly elevated in PDAC." Again, I do not see a critical role of VWF in promoting the clotting cascade, which refers to cleavage activation of clotting factors ultimately leading to thrombin generation and fibrin deposition. 

Ans: We incorporated necessary modification in the revised manuscript.

Section 2.8 is still written in a diffuse, non-structured manner. Why do the authors not clearly separate this section into prevention of VTE (thromboprophylaxis), treatment of established VTE and potential effects of anticoagulation on cancer progression? I am not aware of any guideline recommending apixaban or rivaroxaban for patients who are hospitalized because of malignancy. In the current version of the manuscript, the reference list stops at #76.

Ans: We totally agree and incorporated the suggested subsections under 2.8 in the revised manuscript. We apologize for the truncated reference list you mentioned. We fixed the problem in the revised manuscript.

Reviewer 3 Report

Comments and Suggestions for Authors

The review article by Prouse and colleagues addresses a clinically important topic, that is the high risk of thromboembolic complications associated with pancreatic cancer. Unfortunately, the manuscript lacks a clear structure, and, for the most part, the writing is imprecise and somewhat sloppy. These are just a few examples of the many shortcomings:

1.      Parts of the introduction section are redundant. Balancing prophylactic anticoagulant treatment is irrelevant for the management of established VTE (lines 45 and 46).

2.      Page 3, first paragraph: Plasma does not contain relevant amounts of platelets. Increased thrombin generation in whole blood, but not in plasma, does not exclude a role of hyperresponsive platelets.

3.      What is the critical role of von Willebrand factor (VWF) in promoting the clotting cascade (section 2.3.)? The principal function of VWF is to promote primary, i.e., platelet-dependent hemostasis. Complete lack of VWF results in secondary factor VIII deficiency, but this mechanism is irrelevant for (pancreatic) cancer patients.

4.      I do not think that PDAC tumors secrete preformed thrombin (page 4, first paragraph).

5.      TFPI is the abbreviation for tissue factor pathway inhibitor (page 5, section 2.6.). TFPI inhibits the TF/FVIIa/FXa coagulation initiation complex and has noting to do with plasmin-mediated TF inactivation. This misconception is astonishing for an expert review article on cancer and coagulation.

6.      Patients with pancreatic cancer always have a Khorana score of at least 2 (section 2.7.). Why not simply make this clinically relevant statement?

7.      Section 2.8. on VTE prophylaxis and treatment in pancreatic cancer is simply erroneous. Neither do the authors cite the relevant literature, nor do they clearly differentiate between the various clinical scenarios. This section completely lacks knowledge of anticoagulant treatment strategies and is of no help to clinicians involved in the management of patients with pancreatic cancer.     

Comments on the Quality of English Language

The manuscript needs some improvement in language and grammar.

Author Response

We sincerely appreciate the Reviewer 2 feedback and constructive criticism to strengthen the manuscript.  Here are the point-to-point response to the reviewer’s comments.

Round 2

Reviewer 2 Report

Comments and Suggestions for Authors

The authors provide a clearly improved manuscript.!

They expanded the treatment chapter, thus the title may now be reduced to Pancreatic Cancer and Venous Thromboembolism ! But by doing so they should use updated references such as the ACCP, ASH or ITAC (but 2022!) guideslines and the correct content (all three DXI for LMWH or treatment of CAT)!

Still I am very unhappy by using the term DOAC in the text ! ! For readers not experienced in the field "DOAC" always includes apixaban, edoxaban, dabigatran and rivaroxaban. With regard to cancer patients there are no relevant data for dabigatran in primary prevention or treatment  of VTE. Thus in this group of patients only the direct factor Xa inhibitors (DXI; apixaban, edoxaban, rivaroxaban) should be considered! For all three drugs there are sufficient study data for treatment of cancer associated VTE; but for primary prevention such evidence is only available for apixaban and rivaroxaban! This need to become visible in the text and in figure 2! Using the term DOAC is misleading!

As explained in my former review the Khorana score is not helpful at all in differentiating VTE-risk in patients with pancreatic cancer (this has now been nicely expanded in the manuscript). Thus the Khorana score should not be recommended for these patients as the authors do in figure 2! 

Author Response

We appreciate the reviewer for helpful suggestions and constructive criticism to strengthen the manuscript. Here is our point by point response to reviewer concerns. We highlighted the edited/modified text as well.

the title may now be reduced to Pancreatic Cancer and Venous Thromboembolism

Ans: Done

They expanded the treatment chapter, thus the title may now be reduced to Pancreatic Cancer and Venous Thromboembolism! But by doing so they should use updated references such as the ACCP, ASH or ITAC (but 2022!) guideslines and the correct content (all three DXI for LMWH or treatment of CAT)!

Ans: We tried to incorporate the relevant references and added a few lines to make it more cogent, logical and succinct to read. We also explained why 2019 ITAC guidelines should be adhered to.

Still I am very unhappy by using the term DOAC in the text ! ! For readers not experienced in the field "DOAC" always includes apixaban, edoxaban, dabigatran and rivaroxaban. With regard to cancer patients there are no relevant data for dabigatran in primary prevention or treatment of VTE. Thus in this group of patients only the direct factor Xa inhibitors (DXI; apixaban, edoxaban, rivaroxaban) should be considered! For all three drugs there are sufficient study data for treatment of cancer associated VTE; but for primary prevention such evidence is only available for apixaban and rivaroxaban! This need to become visible in the text and in figure 2! Using the term DOAC is misleading!

Ans: We apologize for our missteps. We tried to correct those portions in text. We also explained at one point unequivocally “There are two broad classes of direct oral anticoagulants (DOACs), one class aims to limit thrombin generation by inhibition of FXa and the other class directly inhibits thrombin proteolytic activity.” We also incorporated the important distinctions “Thus in this group of patients only the direct factor Xa inhibitors (DXI; apixaban, edoxaban, rivaroxaban) should be considered” both in text and in Fig. 2 as suggested by the reviewer.

Reviewer 3 Report

Comments and Suggestions for Authors

The authors have somewhat improved their manuscript. However, the article is still written in a diffuse, non-structured way. Specific issues include the following:

Page 2, line 49: ... such as the JAK-2V617F mutation in myeloproliferative neoplasms (MPN) ...

Page 2, line 52: What does "exacerbates tumor milieu" mean? This phrase is not very precise.

Page 3, first paragraph: "In PDAC, tumor cell-induced platelet aggregation (TCIPA) can promote platelet aggregation via activation of thrombin." This sentence does not make any sense.

Page 3, first paragraph: "VWF is critical in promoting the clotting cascade and is significantly elevated in PDAC." Again, I do not see a critical role of VWF in promoting the clotting cascade, which refers to cleavage activation of clotting factors ultimately leading to thrombin generation and fibrin deposition. 

Section 2.8 is still written in a diffuse, non-structured manner. Why do the authors not clearly separate this section into prevention of VTE (thromboprophylaxis), treatment of established VTE and potential effects of anticoagulation on cancer progression? I am not aware of any guideline recommending apixaban or rivaroxaban for patients who are hospitalized because of malignancy. In the current version of the manuscript, the reference list stops at #76.

Comments on the Quality of English Language

The English language has been improved.

Author Response

(The authors gave the same response as above.)

Round 3

Reviewer 2 Report

Comments and Suggestions for Authors

Modifications sufficiently done ! Still some room for improvement but no reasons to prolong the publication process